# Trefoil factors share a lectin activity that defines their role in mucus

Michael A. Järvå [1,2], James P. Lingford[1,2], Alan John[1,2], Niccolay Madiedo Soler[1,2], Nichollas E. Scott [3] & Ethan D. Goddard-Borger [1,2 ✉]

The mucosal epithelium secretes a host of protective disulfide-rich peptides, including the trefoil factors (TFFs). The TFFs increase the viscoelasticity of the mucosa and promote cell migration, though the molecular mechanisms underlying these functions have remained poorly defined. Here, we demonstrate that all TFFs are divalent lectins that recognise the GlcNAc-α-1,4-Gal disaccharide, which terminates some mucin-like O-glycans. Degradation of this disaccharide by a glycoside hydrolase abrogates TFF binding to mucins. Structural, mutagenic and biophysical data provide insights into how the TFFs recognise this disaccharide and rationalise their ability to modulate the physical properties of mucus across different pH ranges. These data reveal that TFF activity is dependent on the glycosylation state of mucosal glycoproteins and alludes to a lectin function for trefoil domains in other human proteins.

[1] The Walter and Eliza Hall Institute of Medical Research, Parkville, VIC 3052, Australia. [2] Department of Medical Biology, University of Melbourne, Parkville, VIC 3010, Australia. [3] Department of Microbiology and Immunology, University of Melbourne at the Peter Doherty Institute for Infection and Immunity, Parkville, VIC 3010, Australia. ✉email: goddard-borger.e@wehi.edu.au

The three human TFFs (TFF1, TFF2, and TFF3)[1] are ubiquitous in mucosal environments. They share substantial sequence similarity, which is suggestive of a conserved function, though their biological roles are not redundant[2]. The TFFs protect the mucosal epithelium by increasing the viscoelasticity of mucus[3,4] and enhancing epithelial restitution rates[5–9]. TFF overexpression is prevalent in adenocarcinomas[10–12] and a hallmark of chronic inflammatory diseases of the respiratory tract[13–15]. The contribution of TFFs to the progression of these diseases is unclear, although in the context of respiratory diseases any increase in mucus viscoelasticity might be expected to facilitate the formation of obstructions in the airways. Exploring the role of TFFs in these and other biological contexts has been confounded by a poor mechanistic understanding of how they exert their biological activities and a dearth of tools for antagonising TFF activity[2].

The trefoil domain is comprised of three loops (foils) formed by three disulfide bonds (Fig. 1a). TFF1 and TFF3 are disulfide-linked homo-dimers with each monomer possessing a single trefoil domain, while TFF2 is a single-chain protein with two trefoil domains (Fig. 1a). The different ways in which these proteins bring together two trefoil domains impacts the relative orientation, flexibility, and distance between the domains[16–18], which likely contributes to their non-redundant biological activities. An extensive list of TFF binding partners has accumulated within the literature and includes many cell surface and extracellular glycoproteins[2,19], including: β-integrin, CD71, CXCR4/7, FCGBP, DMBT1, GKN2, PAR1/4, LINGO2 and the mucins MUC2, MUC5AC, and MUC6. In the context of the mucus-thickening properties of the TFFs, the soluble mucins MUC5AC and MUC6 are the most relevant binding partners.

Lectin activities have also been reported for some of the TFFs. In the structure of porcine TFF2 a hydrophobic groove was hypothesised to bind glycans[20]. Sometime later it was confirmed that human TFF2 binds α-GlcNAc-terminated mucin-like O-glycans (Fig. 1b)[21]. A lectin-like activity has also been reported for TFF1, which binds *Helicobacter pylori* lipopolysaccharide (LPS) in an α-glucosidase-sensitive manner[22], and gastric mucin[23]. While unique protein–protein interactions may occur between the TFFs and their many reported binding partners[24], a TFF lectin activity would also explain their association with such a diverse array of glycoproteins. However, at present, the lectin activity of each TFF remains poorly characterised. The minimal glycan structure required for TFF2 binding remains ambiguous and its affinity for its cognate ligand is unknown[21]. It also remains unclear if TFF1 binds a similar glycan, since its affinity for *H. pylori* LPS is α-Glc dependent[22]. Furthermore, no data has been reported to suggest that TFF3 is also a lectin.

We sought to definitively address this issue by performing a comprehensive investigation of the lectin activities of all TFFs. Using a combination of ELISA, isothermal titration calorimetry (ITC) and tryptophan quenching assays, we demonstrate that the cognate ligand for all TFFs is the GlcNAc-α-1,4-Gal disaccharide. The binding mode of this disaccharide is revealed using X-ray crystallography and this information used to inform mutagenesis studies to determine which residues are critical for lectin activity. We demonstrate how these residues define the pH profile of lectin activity and how this correlates with the different biological roles of the TFFs. The lectin activity of the TFFs, and the presence of the GlcNAc-α-1,4-Gal disaccharide, is shown to be essential for cross-linking mucus glycoproteins, suggesting that the mucus-thickening properties of the TFFs arise from their ability to reversibly and non-covalently cross-link these large glycoproteins. This information provides a framework for understanding a larger group of hitherto unrecognised mammalian lectins defined by the trefoil domain. Our findings highlight the importance of considering the glycosylation state of mucosal proteins when interpreting the biology of TFFs and is of particular relevance to current and future clinical trials involving the TFFs.

## Results

**α-GlcNAc is required for mucin binding by all TFFs**. To establish that TFF1 and TFF3 possessed the same α-GlcNAc-dependent binding activity as TFF2[21], we prepared site-selectively biotinylated monomeric TFF1 and TFF3 (mTFF1$_{bio}$ and mTFF3$_{bio}$) as well as biotinylated TFF2 (TFF2$_{bio}$) for use as a positive control (Supplementary Table 1). Three mucin samples were also prepared: commercially-available porcine gastric mucin (pMucin), which possesses α-GlcNAc-terminated glycans; reduced and alkylated pMucin (pMucin$_{red}$), which also possesses α-GlcNAc-terminated glycans but lacks tertiary structure[25]; and pMucin$_{red}$ treated with an α-N-acetylglucosaminidase from *Clostridium perfringens* (*Cp*GH89)[26] to remove terminating α-GlcNAc from glycans (pMucin$_{red+GH89}$). ELISA performed using these reagents (Fig. 1c) detected robust binding of the TFF2$_{bio}$ control, mTFF1$_{bio}$, and mTFF3$_{bio}$ to both pMucin and pMucin$_{red}$, while no significant binding was observed for any TFF to pMucin$_{red+GH89}$. This established that all TFF–mucin interactions require α-GlcNAc-terminated glycans and that these interactions are independent of mucin tertiary structure.

**GlcNAc-α-1,4-Gal is necessary and sufficient for TFF binding**. We next sought to determine the minimal carbohydrate structure required for TFF binding and the affinity of this glycan for each TFF. ITC demonstrated that the simple monosaccharides D-GlcNAc, benzyl α-D-GlcNAc and 4-nitrophenyl α-D-GlcNAc had no detectable affinity for TFF1-3. The larger GlcNAc-α-1,4-Gal disaccharide, which is the terminating structure unique to some mucin-like O-glycans, bound to mTFF1, TFF2 and mTFF3 with a $K_d$ of $49 \pm 4\,\mu M$, $44 \pm 8\,\mu M$ and $65 \pm 5\,\mu M$, respectively (Fig. 1d, Supplementary Table 2 and Supplementary Fig. 1). Other α-1,4-linked disaccharides, such as maltose (Glc-α-1,4-Glc) and Gal-α-1,4-Gal, failed to show any affinity for the TFFs by ITC. These results were corroborated by a tryptophan fluorescence quenching assay, which returned $K_d$ values for mTFF1 and mTFF3 of $37 \pm 2\,\mu M$, and $58 \pm 1\,\mu M$, respectively (Fig. 1e).

**Structural insights into disaccharide recognition by TFFs**. Initial attempts to co-crystallise the TFFs and their cognate ligand were confounded by the exceptional solubility of these proteins. Eventually we obtained mTFF1 crystals, though they only yielded an apo structure (Supplementary Table 3 and Supplementary Fig. 2). Following mTFF3 surface lysine methylation, a crystal of the mTFF3–GlcNAc-α-1,4-Gal complex was obtained and the structure determined to a resolution of 1.55 Å (Fig. 2a, b, Supplementary Table 3 and Supplementary Fig. 2). The disaccharide is accommodated within a hydrophobic cleft of TFF3 with ligand binding driven by sterical fitting and hydrogen bonds to the peptide backbone (Fig. 2c). Only two side chains, Asp20 and Trp47, make direct contacts with the disaccharide: Asp20 makes a bidentate hydrogen-bonding interaction between O-4 and O-6 of the non-reducing α-GlcNAc, while C-H–π interactions are made between the indole ring of Trp47 and the reducing-end Gal. Trp47 undergoes considerable motion between liganded and unliganded forms (Fig. 2d, Supplementary Fig. 3). These two residues are conserved in TFF1-3, although TFF1 and TFF2 feature an Asn in place of Asp20 (Fig. 2e). This binding mode is remarkably similar to that of a bacterial carbohydrate-binding module family 32 protein (CBM32), which binds the GlcNAc-α-1,4-Gal glycan ($K_d$ of 72 μM) in the same conformational pose and with analogous intermolecular interactions, despite sharing

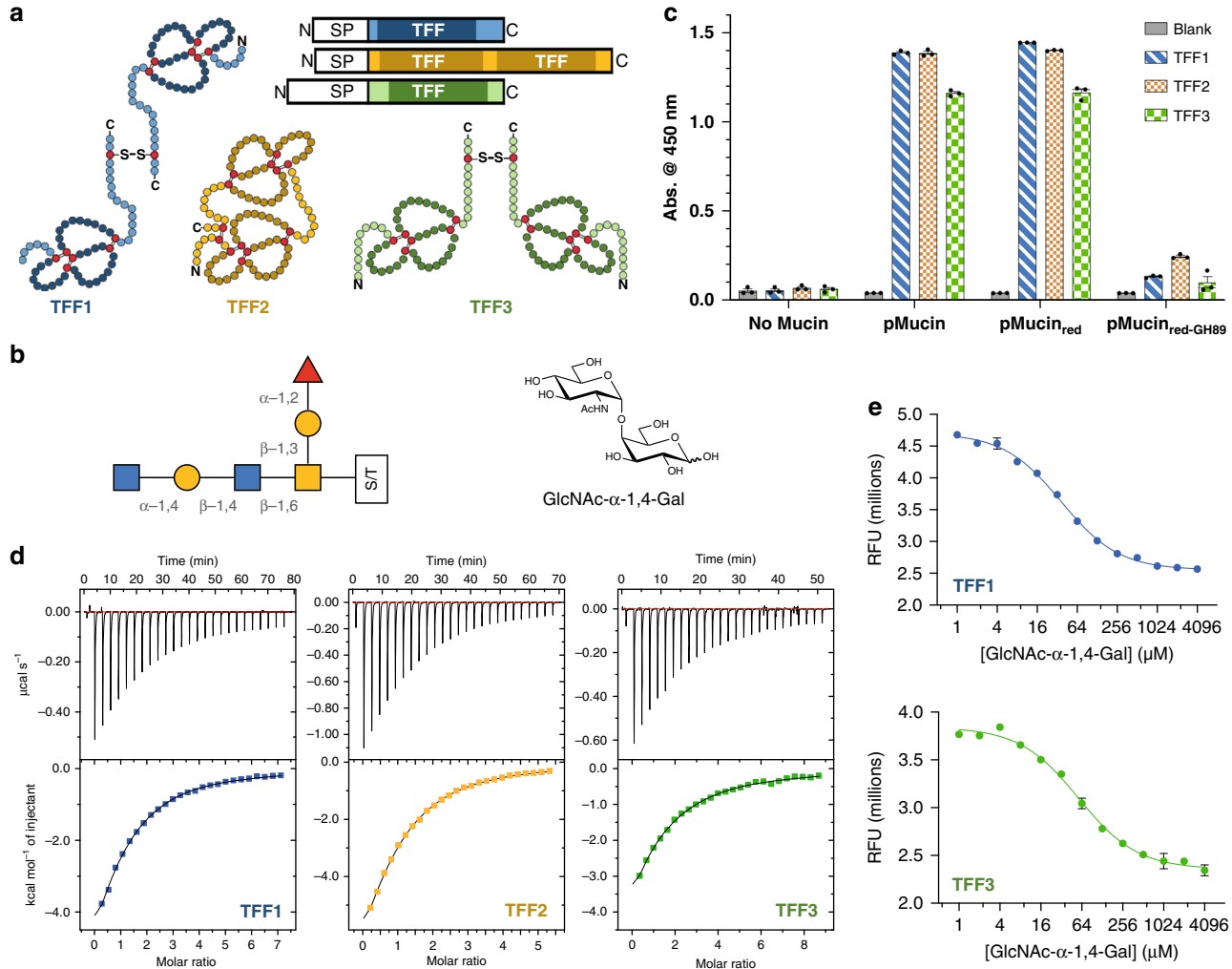

**Fig. 1 All TFFs recognise the GlcNAc-α-1,4-Gal disaccharide. a** Domain architecture and topology of human TFF1-3 (SP = signal peptide). **b** The α-GlcNAc-capped core 2 O-glycan identified as a ligand for TFF2[21]. **c** ELISA data demonstrating the ability of all TFFs to bind to pMucin and pMucin$_{red}$ but not pMucin$_{red+GH89}$. Data are presented as mean values ± SD for three independent replicates. **d** Representative ITC isotherms of mTFF1 (blue), TFF2 (orange), and mTFF3 (green) titrated against the GlcNAc-α-1,4-Gal disaccharide. **e** mTFF1 and mTFF3 binding to GlcNAc-α-1,4-Gal as determined by a tryptophan fluorescence quenching assay. Data are presented as mean values ± SD for three independent replicates. Source data are provided as a Source data file.

no sequence, structural or ancestral commonalities with the TFFs (Fig. 2f, Supplementary Fig. 4)[27].

Mutagenesis of the ligand-binding Asn/Asp and Trp in biotinylated dimeric TFF1 and TFF3 constructs (dTFF1$_{bio}$ and dTFF3$_{bio}$) enabled an examination of the role that these residues play in mucin binding. Using our ELISA, the dTFF1$_{bio}$-N14A mutant was found to have reduced affinity for pMucin$_{red}$ while the dTFF3$_{bio}$-D20A mutant had barely any detectable affinity for pMucin$_{red}$ (Fig. 2a). The dTFF1$_{bio}$-W41A and dTFF3$_{bio}$-W47A mutants were also completely inactive by ELISA. We were unable to detect any binding of GlcNAc-α-1,4-Gal to dTFF1$_{bio}$-N14A or the other TFF mutants by ITC (Supplementary Fig. 1), suggesting that the signal observed for dTFF1$_{bio}$-N14A most likely arises from amplification of a weak residual activity through avidity effects. Indeed, in this assay we have a polyvalent surface (immobilised mucin) interacting with a multivalent binding complex comprised of tetravalent streptavidin-HRP conjugates crosslinked by dimeric TFFs with two biotinylation sites.

**Features that define the pH profile of TFF activity.** An alignment of all mammalian TFF sequences revealed that all TFF1

proteins utilise a disaccharide-binding Asn, while all TFF3 proteins retain an Asp (Supplementary Fig. 5). TFF1 operates in the low pH gastric mucosa, while TFF3 does not, which suggested that a ligand-binding Asn or Asp may impact the pH profile of the lectin activity. To supplement binding data collected at pH 7.4 (Fig. 1e), the $K_d$ for the disaccharide–TFF complexes were determined at pH 5.0 and pH 2.6 using a tryptophan fluorescence quenching assay (Supplementary Fig. 6). At pH 2.6, the $K_d$ for mTFF1 increased only slightly to 67 ± 1 μM, while for mTFF3 the $K_d$ was 350 ± 60 μM; six-fold higher than at pH 7.4. Like TFF1, TFF2 is abundant in gastric mucus, has a conserved Asn (Supplementary Fig. 5) and also binds mucins at low pH[21]. This result suggests that a non-ionisable disaccharide-binding Asn side chain is important for TFF activity at low pH.

**Divalent lectin activity is required for mucin cross-linking.** To demonstrate that the TFFs cross-link soluble mucins in a glycan-dependent manner, we performed agglutination assays using pMucin and our monomeric, dimeric, and mutant TFF1/3 constructs (Fig. 3b, Supplementary Fig. 7). Dimeric TFF1 and TFF3 induced a dose-dependent increase in light scattering over time at

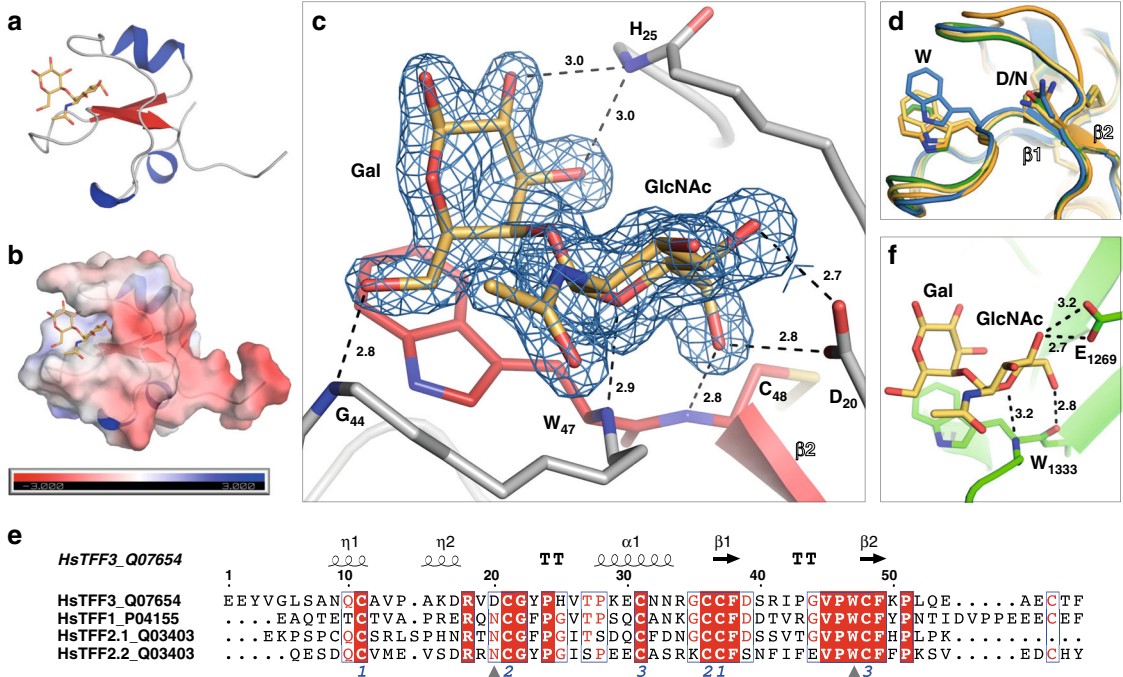

**Fig. 2 Structure of the TFF3:GlcNAc-α-1,4-Gal complex. a** Overall structure of the TFF3:GlcNAc-α-1,4-Gal complex and **b** with an electrostatic surface fitted. **c** Interactions between GlcNAc-α-1,4-Gal and TFF3 (distances in Å) with a Fo-Fc omit map contoured at 1.5σ around the ligand. **d** Superposition of TFF3:GlcNAc-α-1,4-Gal (green), TFF1 (blue), and the trefoil domains from porcine TFF2 (orange and yellow) (PDB ID: 2PSP). The glycan-binding side chains are shown as sticks. **e** Sequence alignment of the trefoil domains in human TFFs. Numbers indicate disulfide connectivity and vertical arrows indicate conserved ligand-binding residues. **f** A CBM32 domain (PDB ID: 4A6O) in complex with GlcNAc-α-1,4-Gal (distances in Å). Source data are provided as a Source data file.

concentrations as low as 4 μM. Monomeric TFFs had no detectable influence on light scattering at concentrations up to 32 μM, commensurate with divalency being required for mucin glycoprotein cross-linking. The mutant dimers, which had greatly diminished or no detectable GlcNAc-α-1,4-Gal disaccharide-binding activity, mirrored the results of the monomers in that they failed to agglutinate pMucins.

**TFF activity is abrogated by a glycoside hydrolase.** We were interested in establishing if the enzymatic degradation of the GlcNAc-α-1,4-Gal structure could liberate TFFs from mucins, since an antagonist of all TFF activity would be a useful tool in the mucosal biology field. ELISAs were used to monitor the liberation of mucin-bound dTFF1bio and dTFF3bio by the *Cp*GH89 enzyme (Fig. 3c). The efficacy of *Cp*GH89 under these conditions were very similar for both TFFs, with EC50 values of 0.8 ± 0.1 nM for TFF1 and 0.4 ± 0.1 nM for TFF3: this is a very effective tool for disrupting mucin–TFF interactions.

**Other mammalian trefoil domains have lectin activities.** It occurred to us that trefoil domains in other mammalian proteins, which are analogous to those in the TFFs, may also have a lectin activity. A phylogenetic analysis of all mammalian proteins with a trefoil domain revealed three clades of putative lectins associated with either amylose-processing enzymes, the GlcNAc-α-1,4-Gal-binding mucosal TFFs studied here in detail, or the glycoprotein-binding zona pellucida proteins (Fig. 4). A re-evaluation of structures available for some of these proteins provided evidence that the trefoil domain of human lysosomal α-glucosidase binds isomaltose (Glc-α-1,6-Glc) (PDB ID: 5KZW) (Supplementary Fig. 8). Biological context suggests that the trefoil domains in sucrase-isomaltase and maltase-glucoamylase likely

bind (iso)maltose, while it is not obvious what ligand the trefoil domains from zona pellucida sperm-binding protein 1 and 4 might bind.

## Discussion

While previous efforts had identified α-GlcNAc-terminated O-glycans as ligands for TFF2[21], it remained unclear what the minimal glycan structure required for binding was, due to the use of ring-opening reductive amination chemistry to prepare neo-glycolipids and the absence of direct biophysical measurements of the TFF–glycan interaction. Here, we have used orthogonal techniques to demonstrate that the cognate ligand for all TFFs is the GlcNAc-α-1,4-Gal disaccharide and that the dissociation constant for these complexes is approximately 50 μM at physiological pH. This relatively weak interaction is typical of many lectins and probably reflects the avidity effects at play in mucus, where each mucin glycoprotein displays many hundreds of O-glycans. Our agglutination assays demonstrated that TFF-mediated aggregation of mucins requires their divalent lectin activity. This supports a model of TFF-modulated mucus rheology where the large mucin glycoprotein polymers are reversibly and non-covalently cross-linked by the TFFs through their α-GlcNAc-terminated O-glycans. This phenomenon would be dominated by avidity effects and necessitates the facile association and dissociation of TFF-mucin complexes to maintain the fluid properties of the mucus.

Our structure of TFF3 in complex with GlcNAc-α-1,4-Gal revealed how these simple proteins recognise their disaccharide ligand using just two residue side chains, and how this approach to binding the disaccharide mimics that of the CBM32 domains, which share no sequence or structural similarities with the TFFs. The structure also provides insights into how TFF1 might bind

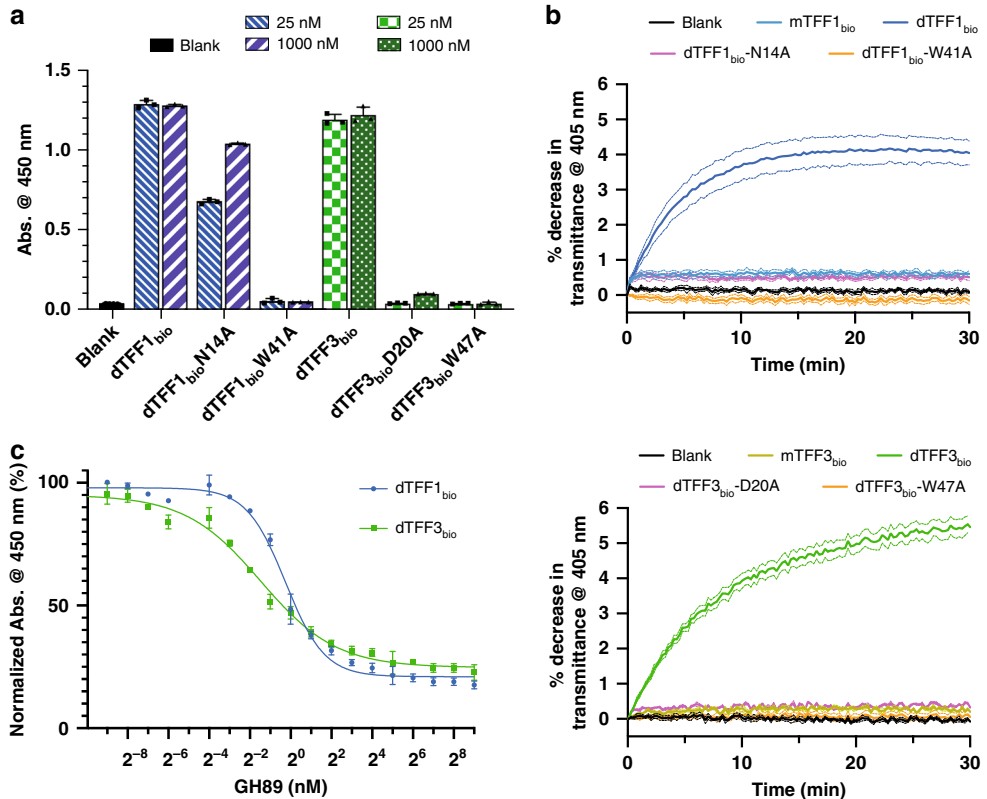

**Fig. 3 Perturbing TFF activity through mutagenesis and glycan degradation. a** Affinity of wild-type and mutant $dTFF1_{bio}$ (blue) and $dTFF3_{bio}$ (green) for $pMucin_{red}$, as determined by ELISA. Data are presented as mean values ±SD for three independent replicates. **b** pMucin agglutination assays using monomeric wild-type, dimeric wild-type or mutant TFF1 (top) and TFF3 (bottom) with optical density at 405 nm monitored with respect to time. **c** $CpGH89$-mediated displacement of dimeric TFF1 (blue circles) and TFF3 (green squares) from immobilised pMucin, as determined by ELISA. Data are presented as mean values ± SD for three independent replicates. Source data are provided as a Source data file.

$H.$ $pylori$ in an α-glucosidase sensitive manner[22]. The core oligosaccharide of many $H.$ $pylori$ strains bears a Glc-α-1,4-Gal-branch[28], while the N-acetyl group of GlcNAc-α-1,4-Gal bound to the TFFs is solvent exposed and makes no interactions with the peptide. Clearly, the Glc-α-1,4-Gal branch of $H.$ $pylori$ core glycans could be easily accommodated in the TFF binding site. We speculate that this unusual $H.$ $pylori$ LPS core oligosaccharide evolved to facilitate the adhesion of this stomach pathogen to the gastric mucosa in a TFF-dependent manner. Evidence that bacteria in mucosal environments have evolved to manipulate mucin-TFF interactions also comes from our observation that recombinant $CpGH89$, and by inference its native variant secreted by $C.$ $perfringens$, is very effective at antagonising TFF activity. This alludes to a role for this enzyme and virulence factor both in scavenging host glycans for sustenance[29] and in disrupting the structural integrity of the mucosa.

Both TFF1 and TFF3 have been the subject of clinical trials: TFF3 was administered to colitis patients[30] and a $Lactococcus$ $lactis$ strain engineered to secrete TFF1 is being administered to patients undergoing chemoradiation therapy to combat oral mucositis (clinicaltrials.gov ID NCT03234465). Our data definitively establishes that the cognate TFF ligand is the GlcNAc-α-1,4-Gal disaccharide and suggests that the other binding partners identified to date, which are all cell-surface and extracellular proteins, may bear mucin-like O-glycans terminated with α-GlcNAc. As such, the presence of the GlcNAc-α-1,4-Gal disaccharide in patient mucus samples could prove to be an important biomarker for predicting responses to the TFF therapies being investigated in the clinic.

While this work has focused on rigorously defining the interactions between TFFs and the soluble mucins, it remains unclear how TFF lectin activities might promote cell migration to achieve epithelial restitution. Conceivably, the divalent TFF lectins could promote the co-localisation of cell-surface glycoproteins like LINGO2, which immunoprecipitates with TFF3[19], to facilitate signalling events that promote cell migration. Assessing the plausibility of this hypothesis is confounded by a paucity of knowledge concerning what proteins bear these unusual α-GlcNAc-terminated O-glycans, and in what biological contexts. These glycans are assembled in the Golgi by α-1,4-N-acetylglucosaminyltransferase (α4GnT)[31,32], which is constitutively expressed only in gastric mucous and Brunner's gland cells[33]. Exploring how α4GnT expression changes in mucosal epithelia in response to inflammatory stimuli, and cataloguing which proteins are modified by this enzyme, are important next steps in understanding the biology of the TFFs.

## Methods

**Production of untagged monomeric TFF1/3.** A series of dsDNA oligonucleotides encoding human TFF1 (UniProt ID: P04155) and TFF3 (UniProt ID: Q07654) with an N-terminal PelB signal peptide, no interchain Cys, and a C-terminal $His_6$-tag codon-harmonised for $E.$ $coli$ (Supplementary Table 4) were synthesised (IDT) and cloned into the pET29b(+) vector (Novagen) using the $NdeI$ and $XhoI$ restriction sites. The resulting plasmids were verified using Sanger sequencing. Each plasmid was transformed into T7 Express cells (NEB) and plated onto LB-agar + 2% glucose + 50 μg ml$^{-1}$ Kan and incubated at 37 °C for 16 h. Single colonies were picked to generate overnight cultures, which were used to inoculate SB media + 0.2% glucose + 50 μg ml$^{-1}$ Kan. The culture was incubated at 37 °C and 220 rpm until it reached an $OD_{600}$ of 1.0. IPTG was added to a final concentration of 0.1 mM for mTFF1 or 0.4 mM for mTFF3 and the culture was incubated for 4 h at

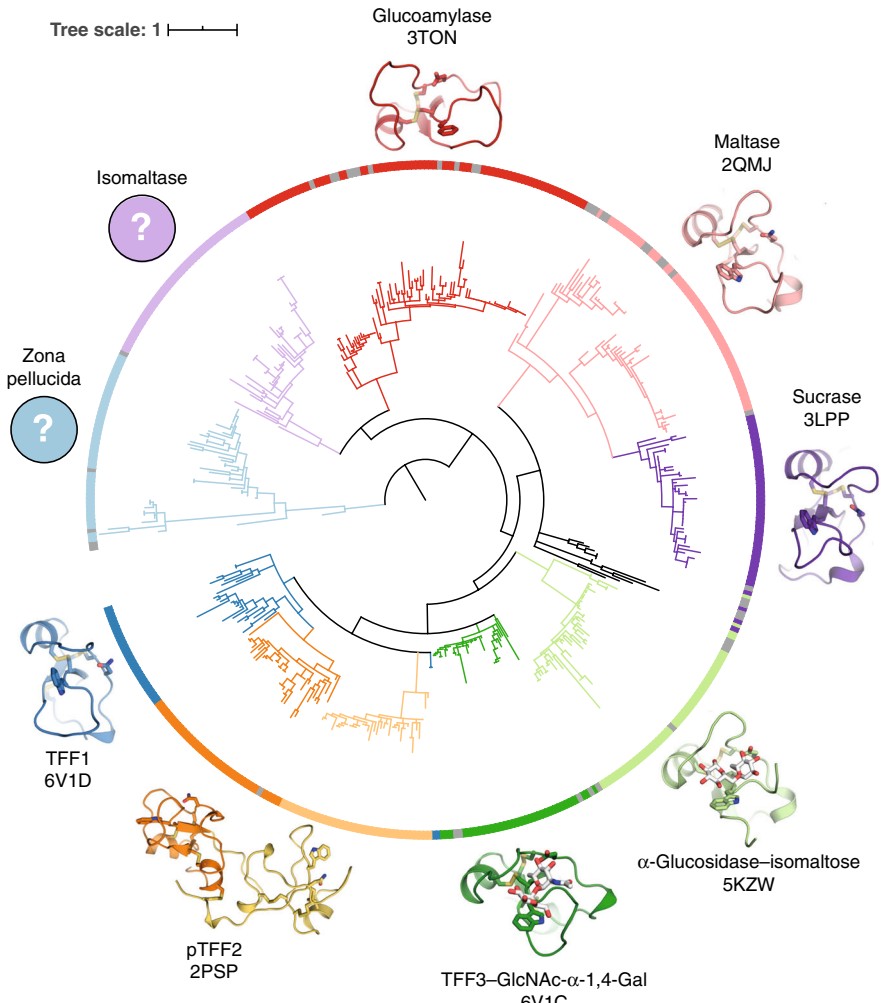

**Fig. 4 A phylogenetic tree of the mammalian trefoil domains.** The large clade of trefoil domains found in amylose-processing enzymes likely bind maltose or isomaltose, as observed for lysosomal α-glucosidase. The clade of TFF trefoil domains bind the GlcNAc-α-1,4-Gal disaccharide found on some mucin-like O-glycans. It is unclear what ligand the third clade, found in zona pellucida sperm-binding protein 1 and 4, might have affinity for.

30 °C and 220 rpm. The cells were harvested by centrifugation (8000×*g*, 25 min, 4 °C) and frozen (−80 °C) until further use. Thawed cells were resuspended in ice-cold high-osmolyte buffer (0.5 M sucrose, 0.2 M Tris, pH 8.0) at 4 °C for 15 min. Three volumes of ice-cold milliQ $H_2O$ was added and the mixture nutated at 4 °C for 30 min. The suspension was adjusted to 150 mM NaCl, 2 mM $MgCl_2$, 20 mM ImH, pH 8.0, centrifuged (18,000×*g*, 30 min, 4 °C) and the supernatant filtered (0.45 μm). The supernatant was applied to a Ni-affinity column (HisTrap Excel, 1 ml, GE Healthcare), the column washed with 10 CV of 50 mM Tris, 300 mM NaCl, 40 mM ImH, pH 7.5, and the protein eluted with 50 mM Tris, 300 mM NaCl, 400 mM ImH, pH 7.5. Fractions containing product, as judged by SDS-PAGE, were pooled and further purified by size exclusion chromatography (Superdex® 75 10/300, GE Healthcare) using 50 mM Tris-HCl, 150 mM NaCl, pH 7.5 as buffer. The C-terminal $His_6$-tag was removed by the addition of 1:100 molar ratio of carboxypeptidase A (Sigma) with incubation at 25 °C for 24 h. The solution was filtered (0.22 μM) and passed through a Ni-affinity column (HisTrap Excel, 1 ml, GE Healthcare) with the flow-through being further purified by size exclusion chromatography (Superdex® 75 10/300, GE Healthcare) using 50 mM HEPES, 50 mM NaCl, pH 7.4 (ITC-Buffer). SDS-PAGE and intact ESI-MS data confirmed the homogeneity of mTFF1 and mTFF3, the removal of the $His_6$-tags and the presence of three disulfide bonds (Supplementary Figs. 9 and 10).

**Production of Avi-tagged-TFF1/3 and mutants thereof.** A series of dsDNA oligonucleotides encoding human TFF1 (UniProt ID: P04155), TFF3 (UniProt ID: Q07654) and mutants thereof with an N-terminal $His_6$-Avi-tag and Factor Xa cleavage site codon-harmonised for *E. coli* (Supplementary Table 4) were synthe-sised (IDT) and cloned into the pET28b(+) vector (Novagen) using the *NcoI* and *XhoI* restriction sites. The resulting plasmids were verified using Sanger

sequencing. Each plasmid was transformed into SHuffle T7 cells (NEB) and plated onto LB-agar + 2% glucose + 50 μg ml$^{-1}$ Kan and incubated at 37 °C for 16 h. Single colonies were picked to generate overnight cultures, which were used to inoculate SB media + 0.2% glucose + 50 μg ml$^{-1}$ Kan. The culture was incubated at 30 °C and 220 rpm until it reached an $OD_{600}$ of 1.0. IPTG was added to a final concentration of 0.4 mM and the culture was incubated for 4 h at 30 °C and 220 rpm. The cells were harvested by centrifugation (8000×*g*, 25 min, 4 °C) and frozen (−80 °C) until further use. The cell pellet from 1.0 l of culture was resuspended in 20 ml ice-cold PBS (50 mM $NaP_i$, 150 mM NaCl, pH 7.5) + 40 mM ImH + 0.5 mM EDTA + 2 μl Benzonaze (Millipore) + 1 tablet of complete EDTA-free protease inhibitor cocktail (Roche). The suspension was sonicated in an ice bath in bursts of 5 s, followed by 10 s pause, at 50% amplitude until total power reached 10 kW. The lysate was centrifuged (18,000×*g*, 30 min, 4 °C) and the supernatant filtered (0.45 μm). The supernatant was applied to a Ni-affinity column (HisTrap Excel, 1 ml, GE Healthcare), the column washed with 10 CV of 50 mM Tris, 300 mM NaCl, 40 mM ImH, pH 7.5, and the protein eluted with 50 mM Tris, 300 mM NaCl, 400 mM ImH, pH 7.5. Fractions containing product, as judged by SDS-PAGE, were pooled and further purified by size exclusion chromatography (Superdex® 75 10/300, GE Healthcare) using 50 mM Tris-HCl, 150 mM NaCl, pH 7.5 as buffer. Non-reducing SDS-PAGE and intact ESI-MS data suggested that these proteins had one free cysteine each: the interchain disulfide had not formed within the cell. These were biotinylated (see below) to generate mTFF1$_{bio}$ and mTFF3$_{bio}$. To make the native dimers with an interchain disulfide bond, protein samples at a concentration of 5 mg ml$^{-1}$ in 100 mM Tris-HCl, pH 8.3 were treated with 20 mM $K_3Fe(CN)_6$ for 16 h at 25 °C. The reaction mixture was filtered (0.22 μm) and purified by size exclusion chromatography (Superdex® 75 10/300, GE Healthcare) using 50 mM Tris-HCl, pH 7.5 as buffer, and anion exchange chromatography (MonoQ 5/50 GL, GE Healthcare) using a gradient over 30 CV from 100% binding buffer (50 mM Tris-HCl, pH 7.5) to 60% elution buffer (binding buffer + 1 M NaCl). SDS-PAGE

and intact ESI-MS data confirmed the homogeneity and number of disulfide bonds in each sample (Supplementary Figs. 9 and 10).

**Production of TFF2.** A dsDNA oligonucleotide encoding human TFF2 (UniProt ID: Q03403) with an N-terminal gp67 signal peptide and a C-terminal His$_{10}$-tag codon-harmonised for *Spodoptera frugiperda* (Supplementary Table 4) was synthesised (IDT) and cloned into the pFastBac vector (ThermoFisher) using the SpeI and XhoI restriction sites. The resulting plasmid was verified using Sanger sequencing. Expression of TFF2 was achieved in *Sf*21 cells using the "Bac-to-Bac Baculovirus Expression System" (ThermoFisher) in accordance with the manufacturer's instructions. Briefly, one litre of cell culture at a density of $1 \times 10^6$ cells ml$^{-1}$ was infected with 30 ml of P3 baculovirus and cultured at 27 °C for 72 h. The culture was centrifuged (8000×*g*, 20 min, 4 °C) and the supernatant collected, adjusted to pH 7.5 and filtered (0.45 µm). The supernatant was applied to a Ni-affinity column (HisTrap Excel, 5 ml, GE Healthcare), the column washed with 10 CV of 50 mM Tris, 300 mM NaCl, 40 mM ImH, pH 7.5, and the protein eluted with 50 mM Tris, 300 mM NaCl, 400 mM ImH, pH 7.5. Fractions containing product, as judged by SDS-PAGE, were pooled and further purified by size exclusion chromatography (Superdex® 75 10/300, GE Healthcare) using 50 mM Tris-HCl, 150 mM NaCl, pH 7.5 as buffer. SDS-PAGE and intact ESI-MS data confirmed that TFF2 was homogenous and had seven disulfide bonds (Supplementary Figs. 9 and 10). A sample of TFF2 was non-specifically biotinylated using EZ-Link™ NHS-PEG$_4$-biotin (ThermoFisher) according to the manufacturer's protocol.

**Biotinylation of Avi-tagged TFFs.** Biotinylation of Avi-tagged TFFs was accomplished by combining 952 µl of 100 µM TFF in PBS, 5 µl of 1 M MgCl$_2$, 20 µl of 100 mM ATP, 20 µl of 40 µM BirA (Sigma), and 3 µl of 50 mM D-biotin. The solution was incubated for 1 h at 30 °C with gentle nutation. An additional 20 µl of 100 mM ATP, 20 µl of 40 µM BirA, and 3 µl of 50 mM D-biotin was added and the solution incubated for an additional hour. The sample was filtered (0.22 µm) and purified by size exclusion chromatography (Superdex® 75 10/300, GE Healthcare) using 50 mM Tris-HCl, 150 mM NaCl, pH 7.5 as buffer. The degree of biotinylation was assessed by intact ESI-MS and was found to be near 100% in all cases (Supplementary Fig. 10).

**Preparation of mucin samples.** A 50 mg ml$^{-1}$ stock of pMucin was prepared by resuspending porcine gastric mucin type III (Sigma) in PBS + 50 mM EDTA + 0.02% NaN$_3$. This pMucin (1 ml) was dialysed (100 kDa NMWL) against 4 M guanidium hydrochloride (2 × 2.0 l) over 48 h at 4 °C. Precipitate was removed by centrifugation (8000×*g*, 20 min, 4 °C) and the soluble supernatant dialysed (100 kDa NMWL) against PBS (2 × 2.0 l) over 48 h at 4 °C. Half of this sample was treated with 50 µg ml$^{-1}$ *Cp*GH89[29] at 30°C for 16 h. Both mucin samples were reduced with 10 mM DTT (30 min, 50 °C) then alkylated by the addition of 30 mM iodoacetamide (30 min, 50 °C). These samples were dialysed (100 kDa NMWL) against PBS (2 × 2.0 l) over 48 h at 4 °C. The concentration of all three mucin stock solutions were standardised to $A_{280}$ = 1.5 using PBS. These samples were aliquoted, flash frozen in liquid N$_2$ and stored at −80 °C.

**TFF-mucin ELISA.** A solution of pMucin, pMucin$_{red}$ or pMucin$_{red+GH89}$ (100 µl, 1 µg ml$^{-1}$) in 100 mM bicarbonate buffer pH 9.4, was added to each well of a 96-well plate (flat-bottom Nunc MaxiSorp, ThermoFisher) and the plate gently nutated for 2 h at 25 °C. The wells were rinsed four times with 200 µl PBS-T (50 mM NaP$_i$ pH 7.4, 150 mM NaCl, 0.2% Tween-20) then blocked for 1 h at 25 °C with blocking buffer (50 mM NaP$_i$ pH 7.4, 150 mM NaCl, 5% w/v BSA). The wells were rinsed four times with 200 µl PBS-T then incubated with 100 µl blocking buffer containing TFF (various concentrations) and 1:1000 Strep-HRP (ThermoFisher) for 1 h at 25 °C. The wells were rinsed four times with 200 µl PBS-T then developed by adding 100 µl of TMB-Turbo (ThermoFisher) and incubated for 30 min at 25 °C. The reaction was quenched by the addition of 100 µl of 2 M H$_2$SO$_4$. Absorbance at 450 nm was read within 20 min using an EnVision 2105 Multimode Plate Reader (PerkinElmer). Data was analysed using Prism 8 (GraphPad).

**Isothermal titration calorimetry.** All TFF and ligand stock solutions were prepared in the same buffer (50 mM HEPES, 50 mM NaCl, pH 7.4) and filtered (0.22 µm) just prior to titration. All control and non-binding ligand experiments were performed as 12 × 3.18 µl injections. For experiments with GlcNAc-α-1,4-Gal (Carbosynth), 25 × 1.58 µl injections were used. All runs were performed at 25 °C in a MicroCal iTC200 (Malvern).

**Tryptophan fluorescence quenching assay.** Tryptophan fluorescence quenching assays were performed at 25 °C on an EnVision 2105 Multimode Plate Reader (PerkinElmer) with excitation at 280 nm and emission measured at 340 nm. The assay was performed in a 384-well plate (Corning, low volume black round bottom polystyrene non-binding surface) using 15 µl per well. Each sample contained mTFF1 or mTFF3 (5 µM) and GlcNAc-α-1,4-Gal (1 µM–4.1 mM) in 50 mM BTP/citric acid, 25 mM NaCl, at either pH 2.6, 5.0, or 7.4. For each ligand dilution series, $K_d$ was calculated by fitting the data to a one-site binding curve using Prism 8 (GraphPad) and the mean calculated from three independent experiments.

**Structure determination of apo-mTFF1.** Sitting drops comprised of 1 µl well solution (1 M ammonium sulfate, 0.1 M Tris-HCl, pH 8.5) and 1 µl mTFF1 solution (10 mg ml$^{-1}$) supplemented with GlcNAc-α-1,4-Gal (5 mM) afforded small clusters of rod-like crystals after one month at 20 °C. The crystals were cryo-protected by supplementing the mother liquor with 3 M ammonium sulfate before being collected on a crystal loop and cryogenically stored in liquid nitrogen. Data was collected on the Australian Synchrotron MX2 beamline[34] at a wavelength of 0.9537 Å and temperature of 100 K, then processed using XDS[35]. The structure was solved by molecular replacement using PHASER[36] and a truncated version of the mTFF3 crystal structure described below as a search model. No density commensurate with a GlcNAc-α-1,4-Gal ligand was observed. The final model of TFF1 in its apo-form was built in Coot[37] and refined with Phenix[38] to a resolution of 2.4 Å. Data collection and refinement statistics are summarised in Supplementary Table 3. The number of reflections for R-free was 338 and values for CC(work), CC(free) and clashscore were 0.927, 0.920 and 0.94, respectively. The Ramachandran plots show that 97.81%, 2.19% and 0% of the amino acids are in favoured, allowed, and disallowed regions, respectively. The coordinates have been deposited in the Protein Data Bank (accession code: 6V1D). Figures were prepared using PyMOL.

**Structure determination of mTFF3:GlcNAc-α-1,4-Gal.** Surface lysine methylation of mTFF3 was conducted using the protocol described by Kim et al.[39] Sitting drops comprised of 0.15 µl well solution (2 M ammonium sulfate, 0.2 M potassium sodium tartrate, and 0.1 M trisodium citrate-citric acid, pH 5.6) and 0.15 µl of permethylated mTFF3 solution (15 mg ml$^{-1}$) supplemented with GlcNAc-α-1,4-Gal (5 mM) afforded a single crystal after 14 days at 20 °C. The crystal was cryo-protected by supplementing the mother liquor with 3 M sodium malonate-malonic acid, pH 5.6, before being collected on a crystal loop and cryogenically stored in liquid nitrogen. Data was collected on the Australian Synchrotron MX2 beamline[34] at a wavelength of 0.9537 Å and temperature of 100 K, then processed using XDS[35]. The structure was solved by molecular replacement using PHASER[36] with a truncated version of the porcine TFF2 crystal structure as a search model (PDB ID: 2PSP)[40]. The final model was built in Coot[37] and refined with Phenix[38] to a resolution of 1.55 Å. Data collection and refinement statistics are summarised in Supplementary Table 3. The number of reflections for R-free was 464 and values for CC(work), CC(free) and clashscore were 0.960, 0.951 and 0, respectively. The Ramachandran plots show that 97.62%, 2.38% and 0% of the amino acids are in favoured, allowed, and disallowed regions, respectively. The coordinates have been deposited in the Protein Data Bank (accession code: 6V1C). Figures were prepared using PyMOL and LigPlot+[41].

**Intact protein MS analysis.** Intact analysis was performed using a 6520 Accurate mass Q-TOF mass spectrometer (Agilent). Protein samples were re-suspended in 2% acetonitrile, 0.1% TFA and 5 µg loaded onto a C5 Jupiter column (5 µm, 300 Å, 50 mm × 2.1 mm, Phenomenex) using an Agilent 1200 series HPLC system. Samples were desalted by washing the column with buffer A (2% acetonitrile, 0.1% formic acid) for 4 min and then separated with a 12 min linear gradient from 2 to 100% buffer B (80% acetonitrile, 0.1% formic acid) at a flow rate of 0.250 ml min$^{-1}$. MS1 mass spectra were acquired at 1 Hz between a mass range of 300–3000 *m/z*. Intact mass analysis and deconvolution was performed using MassHunter B.06.00 (Agilent).

**Mucin agglutination assay.** Mucin agglutination assays were performed at 25 °C on an EnVision 2105 Multimode Plate Reader (PerkinElmer) to monitor transmission at 405 nm over time. The assay was performed in a 96-well plate (Corning, polystyrene flat bottom clear) with 50 µl per well. pMucin concentration was kept consistent at 0.02% w/v and TFF concentration varied (0, 1, 2, 4, 8, 16 or 32 µM). The resulting curves were normalised to percent decrease in signal, averaged over three replicates and plotted using Prism 8 (GraphPad).

**TFF displacement assay.** A solution of pMucin (100 µl, 5 µg ml$^{-1}$) in 100 mM bicarbonate buffer pH 9.4, was added to each well of a 96-well plate (flat-bottom Nunc MaxiSorp, Thermo Scientific) and the plate gently nutated for 2 h at 25 °C. The wells were rinsed four times with 200 µl PBS-T (50 mM NaP$_i$ pH 7.4, 150 mM NaCl, 0.2% Tween-20) then blocked for 1 h at 25 °C with blocking buffer (50 mM NaP$_i$ pH 7.4, 150 mM NaCl, 5% w/v BSA). The wells were rinsed four times with 200 µl PBS-T then incubated with 50 µl blocking buffer containing dTFF1$_{bio}$ or dTFF3$_{bio}$ (16 nM) for 1 h at 25 °C. The wells were rinsed four times with 200 µl PBS-T then treated with 100 µl GH89 (0–512 nM) in blocking buffer for 20 h at 30 °C. The wells were rinsed six times with 200 µl PBS-T then probed with 100 µl Strep-HRP diluted 1:1000 in blocking buffer for 1 h at 25 °C. The wells were rinsed six times with 200 µl PBS-T then developed by adding 100 µl of TMB-Turbo (ThermoFisher) and incubated for 30 min at 25 °C. The reaction was quenched by the addition of 100 µl of 2 M H$_2$SO$_4$. Absorbance at 450 nm was read within 20 min using an EnVision 2105 Multimode Plate Reader (PerkinElmer) and an EC$_{50}$ calculated using Prism 8 (GraphPad) and the mean from three independent experiments.

**Trefoil domain sequence analysis.** An annotated sequence alignment of the trefoil domains of human TFF1, TFF2, and TFF3, was created by ESPript

(esprit.ibcp.fr)[42]. Sequence logos for each of the four mucosal TFF domains in the three human trefoil factors were generated by WebLogo (weblogo.berkeley.edu)[43] using full length protein sequences annotated as mammalian TFF1, TFF2, or TFF3 retrieved from the UniProt database[44] and aligned with Clustal Omega[45]. The phylogenetic tree of all mammalian trefoil domains was created as follows. All trefoil domain sequences corresponding to PF00088 were retrieved aligned according to their hidden Markov model logo generated by the Pfam database[46]. Based on UniProt[44] metadata entries marked as obsolete or truncated were discarded. The remaining sequences were annotated as belonging to one of ten groups (glucoamylase, isomaltase, lysosomal α-glucosidase, maltase, sucrase, TFF1, TFF2-1, TFF2-2, TFF3 or zona pellucida) based on their UniProt annotation. The remaining alignment was used as input to calculate phylogenetic distances with phylogeny.fr[47]. The phylogenetic tree was visualised with iTol[48] and colour-coded according to the described metadata. Where available, a structural representation of each clade of TFF-domains was created using PyMOL and included for eight of the ten groups mentioned above.

**Reporting summary**. Further information on research design is available in the Nature Research Reporting Summary linked to this article.

## Data availability

All relevant data are available from the authors upon request. Structure coordinates have been deposited in the Protein Data Bank (https://www.rcsb.org/) under accession codes PDB 6V1C and PDB 6V1D. The source data underlying Figs. 1c, e, 3a–c and Supplementary Figs. 6, 7 and 9 are provided as a Source data file. All other data are available from the corresponding author on reasonable request.

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

## Acknowledgements

We thank A/Prof. Alisdair Boraston for providing the plasmid for *Cp*GH89 production[29], Dr. Richard Birkinshaw for suggesting surface lysine methylation for protein crystallisation, the beamline staff at the Australian Synchrotron for help with X-ray data collection, Dr. Janet Newman and Dr. Bevan Marshall at the Commonwealth Scientific and Industrial Research Organisation (CSIRO) Collaborative Crystallisation Centre (C3) for assistance in protein crystallisation, and the Melbourne Mass Spectrometry and Proteomics Facility at the University of Melbourne for access to mass spectrometry infrastructure. This research was undertaken in part using the MX2 beamline at the Australian Synchrotron, part of ANSTO, and made use of the Australian Cancer Research Foundation (ACRF) detector. We would like to acknowledge support from: The Walter and Eliza Hall Institute of Medical Research; National Health and Medical Research Council of Australia (NHMRC) project grant GNT1139549; the Australian Cancer Research Fund; and a Victorian State Government Operational Infrastructure support grant.

## Author contributions

M.A.J. performed all biophysical, bioinformatic and structural studies; M.A.J., J.P.L., and A.J. produced recombinant proteins; M.A.J. and N.M.S. performed ELISA assays; N.E.S. performed all mass spectrometry experiments; E.D.G.-B. conceived the project; M.A.J. and E.D.G.-B. co-wrote the manuscript.

## Competing interests

The authors declare no competing interests.
