## [Peer Review File · Nature Communications]

Reviewers' comments:

Reviewer #1 (Remarks to the Author):

Major points

Major concern refers to novelty and priority of claims. Lectin function and specificity of TFF2 were previously reported (J. Biol. Chem. 2014,289: 27363-27375), and implications from this study refer to structural details on both sides, the ligand and its receptor. Due to the structural similarities of TFFs, in particular with respect to the primary sequences in loop 3 (with the highly conserved Trp45 and Phe36 residues) and loop 2, which together form the lectin binding pocket, similar binding characteristics could be expected for TFF1 and TFF3. The latter TFFs are in the focus of the current paper. The work presented is solid and the conclusions drawn are supported by the provided evidence. A further argument against the novelty issue is however provided by authors themselves, as they pre-published the not peer-reviewed paper and uploaded it on Research Gate.

In this context, authors do not adequately refer to the previous work. Reference is made in the appending literature list, but the text refers only to "These observations lie in contrast to a report that TFF2 interacts with an α -DGlcNAc-terminated O-glycan from mucin". The impression is that authors try to hide in a way the merits of the previous publication by concealing that the lectin specificity of TFF2 has been characterized in detail and on a solid physico-chemical level (covering solid-phase binding to neoglycolipids, mass spectrometry and NMR. No further reference is made in the discussion to this paper suggesting to reviewers and readers that the current paper claims priority. On the other hand, authors start the presentation of results with the sentence "To establish that TFF1 and TFF3 possessed the same α -GlcNAc-dependent binding activity as TFF2,...", which means that their work actually extends previous work. What authors can claim actually is the extension of the existing evidence by providing proof of lectin function for other TFFs (TFF1 and TFF3) with similar specificity. These facts need to be clearly stated at prominent position (Abstract, Introduction, Discussion).

p2/3, lines 47-53: Moreover, the statement that the lectin function of TFF2 contrasts with a large number of binding partners is weak, as it ignores possible other mechanisms of interaction. Besides a lectin- carbohydrate interaction, TFFs, like TFF2, could bind to other proteins via hydrophobic interaction. Authors should refer to the paper by Bonar, J. Biol. Chem. 2014, 289: 29677-29690, who provided evidence for the existence of a hydrophobic patch within and at the border of the lectin binding pocket of TFF2.

Also other important references are missing, which point to alternative aspects in host pathogen interactions.

Minor points

There is a need to define terms like “type III mucin-like O-glycans”. This term is not generally used in the field and could refer to blood group structures or to core types or something else.

The claimed minimum structure for binding of TFF lectins was previously reported in JBC 2014 (refer to the solid phase binding assays on neoglycolipids derived from partial structures of the branched mucin glycans. However, it was found to be much less active compared to the trisaccharide GlcNAc α 1-4Gal β 1-4GlcNAc in binding assays on neoglycolipids.

Lectin-mediated binding of TFFs to MUC5AC is questionable, as this mucin has not been shown to express alpha-GlcNAc terminating glycans. The latter is a feature of deep gastric glands in the stomach expressing MUC6.

Franz-Georg Hanisch

Reviewer #2 (Remarks to the Author):

This study by Jarva et al demonstrates that recombinant TFF1, TFF2 and TFF3 all bind the GlcNAc- α -1,4-Gal disaccharide. Using an ELISA binding of monomeric TFF1 and TFF3 to a commercial source of porcine gastric mucin and to the same mucin which had undergone reduction and alkylation but still possesses the α -GlcNAc terminated structure was demonstrated. Treatment of the mucin with an α -D-N-acetylglucosaminidase from *Clostridium perfringens* abrogated binding, thus demonstrating that TFF1 and TFF3 both bind to an α -GlcNAc structure present on mucin similar to what has been reported previously for TFF2 by Hanisch et al. (Ref#20). Isothermal calorimetry and tryptophan fluorescence quenching determined that no binding could be detected to simple monosaccharides but monomeric TFF1 and 3 and TFF2 bound to GlcNAc- α -1,4-Gal disaccharide, a terminating structure found on Type III mucins. Interestingly no binding was detected with other α -1,4 linked disaccharides such as maltose. Crystallisation of the monomeric TFF-GlcNAc- α -1,4-Gal complex indicated that the disaccharide binds within a hydrophobic cleft. Asp20 and Trp47 residues on mTFF3 make direct contact with the glycan. However, in TFF1 and TFF2 an Asn is found in place of Asp 20. Mutagenesis of these Asp/Asn and Trp residues in dimeric forms of TFF1 and TFF3 reduced the affinity of TFF1 for mucin and inhibited completely binding of TFF3. Tryptophan fluorescence quenching assays at different pHs suggest that the conserved Asn20 is important for binding of TFF1 and TFF2 at low pH. The dimeric forms of TFF1 and TFF3 were required for mucin agglutination. Mutant dimers unable to bind GlcNAc- α -1,4-Gal disaccharide were unable to agglutinate mucin. Treatment of mucin containing biotinylated versions of TFF1 and TFF3 with α -D-N-acetylglucosaminidase liberated TFF1 and TFF3 from the mucin. Finally, a phylogenetic analysis of proteins known to have trefoil domains suggests that they may also serve as lectins.

This is a very interesting study on a group of proteins with obvious therapeutic potential, whose mode of action has remained elusive for a long time. Identification of a binding partner for all three trefoil peptides and evidence of how that mediates binding to mucins would be of interest to a large number of investigators and should stimulate further research into these proteins.

Previous studies have shown that the trefoil peptides have lectin activity. Structural studies identified a hydrophobic cleft on TFFs capable of accommodating either an oligosaccharide or an aromatic amino acid side chain (Polshakov et al., 1997, *J. Mol. Biol.* 267: 418–432). Lipopolysaccharide of the gastric pathogen *Helicobacter pylori* was identified as a binding partner for TFF1 (Reeves et al., 2008, Ref #23). More recently Hanisch et al., 2014 (ref# 20), demonstrated that TFF2 bound α -GlcNac and Dunne et al, (*Microorganisms*. 2018 May 18;6(2). pii: E44. doi: 10.3390/microorganisms6020044) reported a striking correlation between binding of TFF1 to human gastric mucins and binding of *Griffonia simplicifolia* lectin-II which is specific for terminal non reducing α - or β - linked N-acetylglucosamine. While some of these references are mentioned in the discussion it would be informative if a short summary of previous work showing that TFFs have lectin activity was clearly stated in the introduction.

The evidence that the recombinant proteins produced for this study bind the GlcNAC- α -1,4-Gal disaccharide is convincing. However, reactivity of the recombinant proteins produced with antibodies raised against different epitopes of the trefoil proteins would be convincing evidence that these proteins are representative of native proteins. If this were possible it would compliment the SDS PAGE results nicely.

While Figure 2a shows that dTFF1_{bio}N14A has reduced affinity for mucin considerable binding still occurs in contrast to binding of dTFF3^{bio}-D20A. This suggests that TFF1 may bind to other glycans present on the mucin. Use of a glycan array to compare binding of TFF1, TFF2 and TFF3 to different glycans could be done to assess if the GlcNAC- α -1,4-Gal disaccharide is the only glycan present on mucins that the TFFs bind to. How specific is the CpGH899 enzyme used to treat the mucin? Is it possible it could cleave other glycans. Considering the site-specific location of TFFs and the association of different TFFs with different mucins it would not be surprising to find that they each may have some unique binding partners.

Have the authors considered the multivalent presentation of glycans on mucins and how important this might be for TFF binding to mucins. Perhaps this could be discussed in the paper.

Reviewer #3 (Remarks to the Author):

In this work, Jarva and colleagues demonstrate that all trefoil factors (TFFs) are divalent lectins that recognize the GlcNAc- α -1,4-Gal disaccharide, which terminates type-III mucin-like O-glycans. The authors report crystal structures of the unliganded form of HsTFF1 and that of HsTFF3 in complex with GlcNAc- α -1,4-Gal, at 2.40 and 1.55 Å resolution, respectively. In combination with mutagenic and biophysical data provided they provide insights into how the TFF peptides recognize the GlcNAc- α -1,4-Gal disaccharide and rationalize their ability to modulate the physical properties of mucus across different pH ranges. In principle, the manuscript is therefore suitable for publication in Nature Communications in terms of its novelty and importance. However, the manuscript can be improved, with several aspects of the work requiring clarifications:

1. page 3, line 60: “To establish that TFF1 and TFF3 possessed the same α -GlcNAc-dependent binding activity as TFF2...” I would suggest to perform the TFF2 binding activities as a control and introduce the results in Figure 1c.
2. page 3, line 63: “reduced and alkylated pMucin (pMucinred), which is denatured...”. Please provide biophysical data or reference in support of this statement.
3. page 4, line 84: Please, prepare two Figures: Figure 1, including panels (a) to (e). Please, increase the size of panels (a), (c), and (d). Figure 2, including panels (f) to (k). Please show the overall structure of TFF1, TFF2 and TFF3. Stereo views. Please, introduce an electrostatic surface representation for the TFF3–GlcNAc- α -1,4-Gal complex. Please, remove panel (h).
4. Table S2. Please, include the following information: (i) reflections used in refinement, (ii) reflections used for R-free, (iii) CC(work), (iv) CC(free), (v) clashcore.
5. page 4, line 93: Please, provide overall superposition of the CBM32 domain (PDB ID: 4A6O) in complex with GlcNAc- α -1,4-Gal with HsTFF3.
6. Can the authors speculate on the molecular mechanism by which TFFs increase the viscoelasticity of the mucosa?

I support publication of the manuscript in the case all these questions can be addressed satisfactorily.

Responses to Reviewer #1

Major points

Major concern refers to novelty and priority of claims. Lectin function and specificity of TFF2 were previously reported (J. Biol. Chem. 2014, 289: 27363-27375), and implications from this study refer to structural details on both sides, the ligand and its receptor.

We don't dispute that lectin activity has been reported for TFF2. Indeed, this reviewer's JBC paper inspired our study and is referenced accordingly in the introduction. To be clear, the JBC paper in question shows that a TFF2 domain recognises glycans with the GlcNAc- α -1,4-Gal disaccharide. It does not show unambiguously that this disaccharide is necessary and sufficient for binding, nor does it quantitate the strength of the TFF2-glycan interaction. Furthermore, it does not explore the ligand preferences of TFF1 or TFF3. The novelties we claim for this paper include: 1) establishing the cognate ligand for TFF1 and TFF3; 2) quantitating the strength of the interaction between all TFFs and their cognate disaccharide ligand using orthogonal biophysical techniques; 3) obtaining the first structural insights into TFF-ligand interactions; 4) demonstrating what TFF residues are important for glycan recognition and how these residues impact the pH profile of their activities, which is relevant to their biological functions; 5) demonstrating that the mucin cross-linking activity of TFFs is dependent on their lectin activity; and 6) identifying for the first time that trefoil domains in other important mammalian proteins may also serve as lectins capable of recognising alpha-linked disaccharides.

Due to the structural similarities of TFFs, in particular with respect to the primary sequences in loop 3 (with the highly conserved Trp45 and Phe36 residues) and loop 2, which together form the lectin binding pocket, similar binding characteristics could be expected for TFF1 and TFF3. The latter TFFs are in the focus of the current paper.

We don't think TFF1 and TFF3 ligand preferences were a forgone conclusion at the outset of this study, especially in light of reports that TFF1 binding to *H. pylori* was alpha-glucosidase dependent (*Gastroenterology*, **2008**, *135*, 2043-2054.e2), as opposed to alpha-hexosaminidase dependent. Also, prior to our structure of the TFF3-disaccharide complex reported here, any discussion of TFF binding pockets was

speculative. Indeed, the conserved Phe36 referred to by the reviewer turns out to play a structural role, while one of the two side chains that interact with ligand turned out to be variable (N or D) and defined the pH activity profile of TFFs.

The work presented is solid and the conclusions drawn are supported by the provided evidence. A further argument against the novelty issue is however provided by authors themselves, as they pre-published the not peer-reviewed paper and uploaded it on Research Gate.

Depositing manuscripts on pre-print servers to rapidly disseminate new knowledge and improve manuscripts through a process of open peer review is the new norm in science and something that is supported by most journals, including *Nat. Commun.*

In this context, authors do not adequately refer to the previous work. Reference is made in the appending literature list, but the text refers only to “These observations lie in contrast to a report that TFF2 interacts with an α -DGlcNAc-terminated O-glycan from mucin”. The impression is that authors try to hide in a way the merits of the previous publication by concealing that the lectin specificity of TFF2 has been characterized in detail and on a solid physico-chemical level (covering solid-phase binding to neoglycolipids, mass spectrometry and NMR. No further reference is made in the discussion to this paper suggesting to reviewers and readers that the current paper claims priority. On the other hand, authors start the presentation of results with the sentence “To establish that TFF1 and TFF3 possessed the same α -GlcNAc-dependent binding activity as TFF2,...”, which means that their work actually extends previous work. What authors can claim actually is the extension of the existing evidence by providing proof of lectin function for other TFFs (TFF1 and TFF3) with similar specificity. These facts need to be clearly stated at prominent position (Abstract, Introduction, Discussion).

Although we certainly didn't intend to ‘hide’ literature precedence for the lectin activity of TFF2, we have made several adjustments to better describe previous contributions. In line with the reviewer's request, we have added references to his paper in the Results and Discussion section, but not in the Abstract as it is prohibited by *Nat. Commun.* editorial policies. We've re-written the introductory paragraph in question and hope the reviewer agrees that it provides a balanced overview of what was and wasn't known about the TFFs prior to our contribution. We would also like to take this opportunity to commend the reviewer's contribution to the field and acknowledge his work as an inspiration for the present manuscript.

p2/3, lines 47-53: Moreover, the statement that the lectin function of TFF2 contrasts with a large number of binding partners is weak, as it ignores possible other mechanisms of interaction. Besides a lectin- carbohydrate interaction, TFFs, like TFF2, could bind to other proteins via hydrophobic interaction. Authors should refer to the paper by Bonar, J. Biol. Chem. 2014, 289: 29677-29690, who provided evidence for the existence of a hydrophobic patch within and at the border of the lectin binding pocket of TFF2. Also other important references are missing, which point to alternative aspects in host pathogen interactions.

We disagree. The evidence that small TFF peptides interact with a broad array of binding partners through different but specific hydrophobic interactions is weak. The hypothesis that TFFs bind to a broad range of glycoprotein partners through their lectin activity is a far more compelling hypothesis, especially in light of the data assembled in this manuscript. Either way, both of these hypotheses remain to be rigorously tested and are presently a matter of opinion rather than fact. That said, in the interest of balance, we have introduced a reference to the reviewer's publication in our introduction. We have also included references to the interaction between TFFs and *H. pylori* in our introduction. These were initially left for the discussion section because we were more focused on the definitive role of mammalian TFFs, rather than their role in host-pathogen interactions.

Minor points

There is a need to define terms like "type III mucin-like O-glycans". This term is not generally used in the field and could refer to blood group structures or to core types or something else.

While this term is used throughout the literature (see *JBC*, **2011**, 286, 6479-89 and *Glycobiology* **2012**, 22, 590-5, among others) we appreciate that some may find this term confusing. As such, we have removed all mention of type-III or class-III mucins.

The claimed minimum structure for binding of TFF lectins was previously reported in *JBC* 2014 (refer to the solid phase binding assays on neoglycolipids derived from partial structures of the branched mucin glycans. However, it was found to be much less active compared to the trisaccharide GlcNAc1-4Galb1-4GlcNAc in binding assays on neoglycolipids.

We assume the reviewer is referring to Figure 9 of his *JBC* paper (*JBC*, **2014**, 289, 27363-75). Using ELISA, he shows that a TFF2 domain binds to a neoglycolipid

derived from the GlcNAc- α -1,4-Gal-b-1,4-GlcNAc trisaccharide, with barely detectable activity for a neoglycolipid derived from the GlcNAc- α -1,4-Gal disaccharide. In retrospect, he probably observed very weak binding for the disaccharide because neoglycolipid formation through reductive amination ring-opens the reducing-end sugar. The unknown impact of this chemical modifications on binding is why the minimum structural motif required for TFF2 binding was ambiguous. We have eliminated this ambiguity by collecting biophysical data using unmodified glycans and full-length TFF2. We maintain that our manuscript is the first to definitely establish the cognate ligand for TFF2 and that all of our results for TFF1 and TFF3 are entirely novel.

Lectin-mediated binding of TFFs to MUC5AC is questionable, as this mucin has not been shown to express alpha-GlcNAc terminating glycans. The latter is a feature of deep gastric glands in the stomach expressing MUC6.

We respectfully disagree. TFF1 was shown to bind to MUC5AC quite some time ago (*Cell. Mol. Life Sci.* **2004**, *61*, 1946–1954) and in the last few years α -GlcNAc-terminated glycans have been detected on MUC5AC (*PLoS One* **2016**, *11* (12), e0167070). Knowledge of which mucosal proteins possess the terminating α -GlcNAc glycan is limited because no comprehensive glycoproteomic survey has ever been conducted to look specifically at this modification. In our opinion, it would be remarkable if A4GNT, a Golgi enzyme, didn't modify many other cell surface or secreted glycoproteins. We hope that our work will provide the impetus required to perform such investigations, which will no doubt help address some of the questions debated here.

Responses to Reviewer #2

This study by Jarva et al. demonstrates that recombinant TFF1, TFF2 and TFF3 all bind the GlcNAc- α -1,4-Gal disaccharide. Using an ELISA binding of monomeric TFF1 and TFF3 to a commercial source of porcine gastric mucin and to the same mucin which had undergone reduction and alkylation but still possesses the α -GlcNAc terminated structure was demonstrated. Treatment of the mucin with an α -D-N-acetylglucosaminidase from *Clostridium perfringens* abrogated binding, thus demonstrating that TFF1 and TFF3 both bind to an α -GlcNAc structure present on mucin similar to what has been reported previously for TFF2 by Hanisch et al. (Ref#20). Isothermal calorimetry and tryptophan fluorescence quenching determined that no binding could be detected to simple monosaccharides but monomeric TFF1 and 3 and TFF2 bound to GlcNAc- α -1,4-Gal disaccharide, a terminating structure found on Type III mucins. Interestingly no binding was detected with other α -1,4 linked disaccharides such as maltose.

Crystallisation of the monomeric TFF-GlcNAc- α -1,4-Gal complex indicated that the disaccharide binds within a hydrophobic cleft. Asp20 and Trp47 residues on mTFF3 make direct contact with the glycan. However, in TFF1 and TFF2 an Asn is found in place of Asp 20. Mutagenesis of these Asp/Asn and Trp residues in dimeric forms of TFF1 and TFF3 reduced the affinity of TFF1 for mucin and inhibited completely binding of TFF3. Tryptophan fluorescence quenching assays at different pHs suggest that the conserved Asn20 is important for binding of TFF1 and TFF2 at low pH. The dimeric forms of TFF1 and TFF3 were required for mucin agglutination. Mutant dimers unable to bind GlcNAc- α -1,4-Gal disaccharide were unable to agglutinate mucin. Treatment of mucin containing biotinylated versions of TFF1 and TFF3 with α -D-N-acetylglucosaminidase liberated TFF1 and TFF3 from the mucin. Finally, a phylogenetic analysis of proteins known to have trefoil domains suggests that they may also serve as lectins.

This is a very interesting study on a group of proteins with obvious therapeutic potential, whose mode of action has remained elusive for a long time. Identification of a binding partner for all three trefoil peptides and evidence of how that mediates binding to mucins would be of interest to a large number of investigators and should stimulate further research into these proteins.

Previous studies have shown that the trefoil peptides have lectin activity. Structural studies

identified a hydrophobic cleft on TFFs capable of accommodating either an oligosaccharide or an aromatic amino acid side chain (Polshakov et al., 1997, J. Mol. Biol. 267: 418–432). Lipopolysaccharide of the gastric pathogen *Helicobacter pylori* was identified as a binding partner for TFF1 (Reeves et al., 2008, Ref #23). More recently Hanisch et al., 2014 (ref# 20), demonstrated that TFF2 bound α -GlcNAc and Dunne et al, (Microorganisms. 2018 May 18;6(2). pii: E44. doi: 10.3390/microorganisms6020044) reported a striking correlation between binding of TFF1 to human gastric mucins and binding of Griffonia simplicifolia lectin-II which is specific for terminal non reducing α - or β - linked N-acetylglucosamine. While some of these references are mentioned in the discussion it would be informative if a short summary of previous work showing that TFFs have lectin activity was clearly stated in the introduction.

We have expanded our discussion of the known TFF activities in the Introduction. We hope the reviewer agrees that it is a fair and balanced overview of the field.

The evidence that the recombinant proteins produced for this study bind the GlcNAc- α -1,4-Gal disaccharide is convincing. However, reactivity of the recombinant proteins produced with antibodies raised against different epitopes of the trefoil proteins would be convincing evidence that these proteins are representative of native proteins. If this were possible it would complement the SDS PAGE results nicely.

The homogeneity of our proteins is demonstrated by SDS-PAGE after gel filtration chromatography (Supplementary Figure 9) and intact ESI-MS (Supplementary Figure 10). The latter also demonstrates that all disulfide bonds are formed. The native fold of our proteins is further evident from our crystal structures of TFF1 and TFF3, and the activity assays (ELISA, ITC, Trp fluorescence, agglutination) conducted for TFF1, TFF2 and TFF3. We believe this provides an extremely high degree of confidence that these proteins possess their native fold. The reactivity of antibodies against our proteins is not diagnostic of native fold unless said antibodies have a well-defined conformational epitope: unfortunately, no such antibodies with this level of validation are presently available for the TFFs.

While Figure 2a shows that dTFF1bioN14A has reduced affinity for mucin considerable binding still occurs in contrast to binding of dTFF3bio-D20A. This suggests that TFF1 may bind to other glycans present on the mucin. Use of a glycan array to compare binding of TFF1, TFF2 and TFF3 to different glycans could be done to assess if the GlcNAc- α -1,4-Gal

disaccharide is the only glycan present on mucins that the TFFs bind to. How specific is the CpGH89 enzyme used to treat the mucin? Is it possible it could cleave other glycans. Considering the site-specific location of TFFs and the association of different TFFs with different mucins it would not be surprising to find that they each may have some unique binding partners.

CpGH89 is highly specific for the GlcNAc- α -1,4-Gal disaccharide (*J. Biol. Chem.* **2011**, *286*, 6479-89): it does not cleave other mammalian α -GlcNAc structures (GlcNAc- α -1,2/3/6-Gal or GlcNAc- α -1,4-GlcA). This information, combined with the observation that WT dTFF1bio does not bind to GH89-treated mucin, suggests that TFF1 has a high specificity for the GlcNAc- α -1,4-Gal disaccharide and does not bind other mucin glycan structures. We believe that the ELISA signal observed for dTFF1bioN14A is a manifestation of avidity effects that amplify weak residual binding of the mutant. While we could not detect any residual binding for dTFF1bioN14A to GlcNAc- α -1,4-Gal by ITC, the ELISA assay involves a polyvalent surface (immobilised mucin) and a multivalent binder (dimeric dTFF1bioN14A with two biotins crosslinking tetravalent streptavidin-HRP conjugates). As such, we think the most likely explanation for this residual binding by ELISA is avidity-mediated amplification of otherwise very weak binding by the dTFF1bioN14A mutant. We have amended the manuscript text to better explain this point.

Have the authors considered the multivalent presentation of glycans on mucins and how important this might be for TFF binding to mucins. Perhaps this could be discussed in the paper.

Absolutely! Avidity plays a defining role in the interactions between the divalent TFFs and polyvalent mucins. The weak ($K_d \approx 50 \mu\text{M}$) affinity of TFFs for their cognate ligand likely ensures rapid equilibration to maintain a dynamic glycoprotein network. A short paragraph to better explain our thoughts on this effect has been added to the Discussion.

Responses to Reviewer #3

In this work, Jarva and colleagues demonstrate that all trefoil factors (TFFs) are divalent lectins that recognize the GlcNAc- α -1,4-Gal disaccharide, which terminates type-III mucin-like O-glycans. The authors report crystal structures of the unliganded form of HsTFF1 and that of HsTFF3 in complex with GlcNAc- α -1,4-Gal, at 2.40 and 1.55 Å resolution, respectively. In combination with mutagenic and biophysical data they provide insights into how the TFF peptides recognize the GlcNAc- α -1,4-Gal disaccharide and rationalize their ability to modulate the physical properties of mucus across different pH ranges. In principle, the manuscript is therefore suitable for publication in Nature Communications in terms of its novelty and importance. However, the manuscript can be improved, with several aspects of the work requiring clarifications:

1. page 3, line 60: “To establish that TFF1 and TFF3 possessed the same α -GlcNAc-dependent binding activity as TFF2...” I would suggest to perform the TFF2 binding activities as a control and introduce the results in Figure 1c.

This is a great suggestion. We have biotinylated our recombinant human TFF2 and repeated the ELISA experiment in Figure 1c with all three TFFs.

2. page 3, line 63: “reduced and alkylated pMucin (pMucinred), which is denatured...”.

Please provide biophysical data or reference in support of this statement.

We have included a relevant reference for the denaturation of porcine gastric mucin – this is a standard technique.

3. page 4, line 84: Please, prepare two Figures: Figure 1, including panels (a) to (e). Please, increase the size of panels (a), (c), and (d). Figure 2, including panels (f) to (k). Please show the overall structure of TFF1, TFF2 and TFF3. Stereo views. Please, introduce an electrostatic surface representation for the TFF3–GlcNAc- α -1,4-Gal complex. Please, remove panel (h).

We have taken the reviewer’s advice and prepared two figures from Figure 1. The new Figure 1 includes every change requested by the reviewer. The new figure 2 includes all of the requested changes except the stereoview of TFF1-3, which we have placed in the supplementary information (Supplementary Figure 4).

4. Table S3. Please, include the following information: (i) reflections used in refinement, (ii) reflections used for R-free, (iii) CC(work), (iv) CC(free), (v) clashcore.

The editor has asked us to use the *Nature Research* template to report the X-ray crystallography refinement statistics. As such, Supplementary Table 3 has changed a little and includes the number of reflections used in refinement. To satisfy the reviewer's request we have added the number of reflections used for R-free and values for CC(work), CC(free) and clashscore to the methods section.

5. page 4, line 93: Please, provide overall superposition of the CBM32 domain (PDB ID: 4A6O) in complex with GlcNAc- α -1,4-Gal with HsTFF3.

An overall superposition of CBM32 and TFF3 aligned through their bound GlcNAc- α -1,4-Gal ligand is now available in Supplementary Figure 5.

6. Can the authors speculate on the molecular mechanism by which TFFs increase the viscoelasticity of the mucosa?

We speculate that the TFFs modulate mucus viscosity by reversibly and non-covalently cross-linking mucin glycoprotein polymers through their α -GlcNAc-terminated O-glycans. This increases the force required for mucin polymers to flow past each other in solution. Our manuscript has identified and validated CpGH89 as a useful tool for testing this hypothesis in mucus samples, though such a study goes beyond the scope of the present manuscript. A short paragraph to this effect has been added to the Discussion.